# Genomic and Morphological Differentiation of Spirit Producing *Agave angustifolia* Traditional Landraces Cultivated in Jalisco, Mexico

**DOI:** 10.3390/plants11172274

**Published:** 2022-08-31

**Authors:** Dánae Cabrera-Toledo, Eddy Mendoza-Galindo, Nerea Larranaga, Alfredo Herrera-Estrella, Marilyn Vásquez-Cruz, Tania Hernández-Hernández

**Affiliations:** 1Laboratorio Nacional de Identificación y Caracterización Vegetal (LaniVeg), Departamento de Botánica y Zoología, Universidad de Guadalajara, Zapopan 45200, Mexico; 2Agrogenomic Sciences, ENES Leon UNAM, Guanajuato 37689, Mexico; 3LANGEBIO-UGA, Guanajuato 36821, Mexico; 4Research and Collections, Desert Botanical Garden, Phoenix, AZ 85008, USA

**Keywords:** *A. angustifolia*, *A. tequilana*, *A. rhodacantha*, tequila, mezcal, mescal, *Agave*

## Abstract

Traditional agave spirits such as mezcal or tequila are produced all over Mexico using different species of *Agave*. Amongst them, *A. angustifolia* is the most popular given its agricultural extension. *A. angustifolia* is a wild species extensively distributed from North to Central America, and previous studies show that it is highly related to the tequila agave *A. tequilana*. In different regions of Mexico, *A. angustifolia* is cultivated under different types and levels of management, and although traditional producers identify several landraces, for the non-trained eye there are no perceivable differences. After interviews with producers from different localities in Jalisco, Mexico, we sampled *A. angustifolia* plants classified as different landraces, measured several morphological traits, and characterized their genetic differentiation and diversity at the genome-wide level. We included additional samples identified as *A. tequilana* and *A. rhodacantha* to evaluate their relationship with *A. angustifolia*. In contrast with previous studies, our pool of ca 20K high quality unlinked SNPs provided more information and helped us to distinguish different genetic groups that are congruent with the ethnobotanical landraces. We found no evidence to genetically delimitate *A. tequilana*, *A. rhodacantha* and *A. angustifolia*. Our large genome level dataset allows a better understanding of the genetic identity of important *A. angustifolia* traditional and autochthonous landraces.

## 1. Introduction

The use of *Agave* for spirits production is recent in human history. Although it has been suggested that *Agave* plants have been used as a source of food or fiber production for at least 9000 years [1,2], the distillation of spirits using *Agave* did not begin until the 16th century and the cultivation and production of spirits increased with the popularity of tequila as recently as the early 19th century [3,4,5,6,7,8]. This indicates the extremely recent initiation of the management and domestication processes of *Agave* with spirit production purposes, especially for some landraces such as the tequila plant. This, together with the long life-cycle particular of agaves, explains the incipient domestication status of cultivars and landraces.

From all spirits produced in Mexico using *Agave*, tequila is the most emblematic. Made from the “blue agave” plant *Agave tequilana* var. “Azul”, tequila has been produced traditionally since the 18th century in the state of Jalisco (Figure 1). After the Denomination of Origin of Tequila (DOT) obtained in 1977 [9], the number of plants cultivated for tequila production has increased enormously [10]. The blue agave is now massively cultivated and used in a highly industrialized system owned by transnational corporations [11]. Unfortunately, this industrialization also brought expensive social and environmental consequences, including the erosion of the diversity of traditional landraces [11,12,13,14,15]. Since the DOT established that only blue agave can be used to make tequila, producers shifted their efforts to grow only blue agave landrace, severely mining the diversity of landraces that used to be part of traditional tequila agroecosystems 400 years ago [8,11,16]. In addition, to preserve the integrity of the blue agave, producers have historically recurred almost exclusively to asexual propagation, a strategy that has also been adopted for the cultivation of other landraces, resulting in an extreme loss of genetic diversity [17,18]. These current conditions urge the development and promotion of traditional and new landraces locally adapted to regional climatic regimes and conditions [11,14,15,19,20].

Mexico is the center of the diversity and domestication of *Agave* [21], where 53 species are used for spirit production, most widely known as “mezcal” [22,23]. However, *A. angustifolia* is the most extensively distributed and cultivated for this purpose. *A. tequilana, A. angustifolia* and *A. rhodacantha* belong to a morphological and genetic complex of species known as *A. angustifolia* complex [24,25]. Previous studies using traditional molecular markers have proposed wild populations of *A. angustifolia* in the state of Jalisco as possible wild closest relatives of the blue agave [17,18,26]. Ethnobotanical evidence shows that Southern Jalisco is the nucleus of the greatest diversity of spirit producing landraces within the *A. angustifolia* complex, and it is likely that this area was where traditional farmers initiated the selection of *Agave* germplasm for spirit production [8]. Traditional mezcal-producing farmsteads in this region are known as “mezcaleras” and currently sustain around 20 landraces [18,24]. Another important region in the state of Jalisco where mezcal is traditionally produced is the North Coast, but, until now, ethnobotanical documentation in this region has not been performed and landraces have not been studied (Figure 1). In the North Coast, producers recognize about 10 different landraces from the same *A. angustifolia* complex (Huerta-Galván, unpublished data). As it happened with tequila, mezcal has also gained worldwide recognition and explosive demand. Over the period between 2011 and 2019, mezcal production in Mexico increased over 700% [27], with the US being the highest buyer of this spirit around the world [28]. Under this increasing pressure, it is important to document, understand, and preserve alternative autochthonous landraces and traditional agroecosystems, avoiding the adoption of procedures such as massive, industrialized monoculture by clonal reproduction as is the case with the blue agave.

Although some species and landraces included in the *A. angustifolia* complex have been studied using traditional genetic markers such as AFLPs and microsatellites [18,24,25,26], none of the Jalisco traditional landraces have been evaluated using a genomic approach with next generation sequencing techniques. The implementation of a low-cost, high-throughput genotyping-by-sequencing (GBS) approach allows the identification of thousands of markers per sample [29], and it is a method particularly useful when working with species with large genomes such as *Agave* [30]. In this study, we implement GBS to evaluate if there is a congruence amongst the ethnobotanical description, morphology, and genetic association of wild, semi-wild and established landraces within the *A. angustifolia* complex in Jalisco. We visited 12 mezcaleras both in the North Coast and in Southern Jalisco and interviewed with local producers. We sampled several individual plants that were referred to by them as different landraces or semi-wild landraces (“Cimarrón”, “Sierreño” or “Barranqueño”). We also sampled *A. angustifolia* plants in the wild, as well as additional *A. tequilana* and *A. rhodacantha* accessions, to evaluate their genetic identity and characterize their genetic diversity in relation to geography and ethnobotanical characterization. Finally, we conducted several morphological measurements on a subset of our cultivated sampled plants that were classified by producers as belonging to the widest recognized landraces. This study contributes with the largest available dataset at the genomic level for local Jalisco traditional landraces. Our pool of SNPs allowed us to better understand the genetics underneath the cultivation, propagation and domestication processes of species, varieties, and landraces within the *A. angustifolia* complex in the region.

## 2. Results

### 2.1. Sampling and SNP Calling

In total, we collected 87 samples in the state of Jalisco, Mexico between 2020 and 2021 (Figure 1 and Figure 2 and Table 1). With our sequencing approach, we generated about 5 million reads per sample with a mean per-site depth of 6.59X. Our SNP calling procedure using a reference genome of *A. tequilana* var. “Azul” allowed us to generate a matrix of 67,175 high-quality SNPs, with a mean individual SNP coverage of 0.91%. Linked SNPs within a window size of 50, a step of 10 and square-r threshold of 0.1 were pruned, leaving a final matrix of 19,983 SNPs after filtering, suggesting about 70% of the original SNP dataset was linked. According to this dataset, the *A. angustifolia* Jalisco complex studied with our samples showed low genetic variation, with a mean expected heterozygosity value (*H_e_*) of 0.1311 ± 0.0110 and mean nucleotide diversity (***π***) of 0.2622 ± 0.0220.

### 2.2. Genetic Differentiation and Population Structure

The pairwise *F_ST_* mean value (0.3747 ± 0.0740) showed moderate-to-high genetic divergence across individuals. To further explore the population structure, we carried out an Identity by State (IBS) analysis to compare pairwise genetic distances. We did not find a full congruence of the clusters generated using genetic distances with the landraces. To verify, we conducted an Identity by Descent (IBD) test. Without exception, all pairwise IBD values suggested each sample is genetically related to each other. Therefore, we omitted genetic distances in the following procedures.

We carried out a Principal Component Analysis (PCA) using our filtered SNP dataset. The first three components of the PCA explained 13.9321%, 11.1120% and 10.4036% of the variance, respectively. The Euclidean distances of the eigenvectors from the three components were used to build a UPGMA genetic dendrogram (Figure 3). Four major groups were clearly defined, with observed heterozygosity values, while expected heterozygosity values are lower in the groups including a single landrace (Table 2). The first and most divergent one corresponds to what we call Cluster 1, including all the “Amarillo” landrace samples coming from four different mezcaleras in Cabo Corrientes (North Coast; localities and samples information in Table 1). The next group corresponds to what we designate as Cluster 2, with samples from Southern Jalisco. Cluster 2 includes all the plants sampled in Tolimán (Southern Jalisco) that were classified by local producers as “Lineño” landrace. According to our results, these samples group together in a close relationship with two genetically similar individuals classified one as “Cimarrón” (coming from the same farmstead in Tolimán) and another wild plant coming from a nearby area (AP2 from Apango, Jalisco, 30 km northeast from Tolimán). The following group is what we call Cluster 3 and includes two genetically separated subgroups: 3a and 3b. Subcluster 3a includes several samples that are related by producers to similar types: “Ixtero Amarillo” from Southern Jalisco, with samples from multiple mezcaleras in different locations (IALA1, IALA10 from Zapotitlán de Vadillo, IAMG from Tolimán, TET8 and TET11 from Tetapán,) and “Ixtero Amarillo Barranqueño” landrace (TOL8) from Tolimán. Subcluster 3a is highly related to 3b, which includes a mixture of different landraces, as well as a wild plant from Southern Jalisco: “Ixtero Amarillo” (IAMG1 from Tolimán, TET10 from Tetiapán, IATE2 from Zapotitlán), “Lineño” (TET7 from Tetiapán, TOL9 from Tolimán), “Cimarron” (TET3 and TET1 from Tetiapán), as well as a wild plant from Apango (AP5).

**Table 1 plants-11-02274-t001:** Samples and information in this study. Municipality or Locality name where collected, species name, landrace according to producer, code used in this study, voucher number if available, and number of accessions (*N*).

Municipality or Locality	Species	Landrace	Code	Voucher	*N*
Cabo Corrientes	*A. angustifolia*	Amarillo	LCS1, LCS10	MPDM-218	2
Cabo Corrientes	*A. angustifolia*	Amarillo	PDSA1, PSDA10	MPDM-243	2
Cabo Corrientes	*A. angustifolia*	Amarillo	NJS10	-	1
Cabo Corrientes	*A. angustifolia*	Amarillo	ARR1	MPDM-215	1
Tolimán	*A. angustifolia*	Ixtero Amarillo Barranqueño	TOL8	-	1
Tequila	*A. tequilana*	Azul	CIAT1-2	-	2
Tolimán	*A. tequilana*	Azul	TOL13	-	1
Tolimán	*A. rhodacantha*	Azul Telcruz	SOMG1	-	1
Zapotitlán de Vadillo	*A. rhodacantha*	Azul Telcruz	ATLA9	MPDM-273	1
Cabo Corrientes	*A. rhodacantha*	Cenizo	CJJ1, CJJ3, CJJ5	MPDM-219	3
Cabo Corrientes	*A. rhodacantha*	Cenizo	PBS5	MDPM-223	1
Tequila	*A. tequilana*	Chato	CIAH1-2	-	2
Cabo Corrientes	*A. rhodacantha*	Chico Aguiar	CHJJ3	PCR-9688	1
Cabo Corrientes	*A. rhodacantha*	Chico Aguiar	DVD1, DVD7	MPDM-235	2
Zapotitlán de Vadillo (Tetiapán)	*A. angustifolia*	Cimarrón	TET1-3	-	3
Tolimán	*A. angustifolia*	Cimarrón	TOL1-4	-	4
Tolimán	*A. angustifolia*	Cimarrón Verde	CVMG	-	1
Zapotitlán de Vadillo (Tetiapán)	*A. rhotacantha*	Ixtero Amarillo	TET8-11	-	4
Tolimán	*A. rhodacantha*	Ixtero Amarillo	IAMG1	-	1
Zapotitlán de Vadillo	*A. rhodacantha*	Ixtero Amarillo	IALA1, IALA10	MPDM-271	2
Zapotitlán de Vadillo	*A. rhodacantha*	Ixtero Amarillo	IATE1	-	1
Zapotitlán de Vadillo	*A. aff. rhodacantha*	Ixtero Verde	IVLA3	MPDM-269	1
Zapotitlán de Vadillo (Tetiapán)	*A. angustifolia*	Lineño	TET4-7	-	4
Tolimán	*A. angustifolia*	Lineño	TOL5-6	DCT-23, DCT-25, DCT-26	3
Tolimán	*A. angustifolia*	Lineño Ixtero	TOL7	-	1
Tolimán	*A. angustifolia*	Lineño silvestre	TOL9-12	-	4
Tolimán	*A. angustifolia*	Negro Cimarrón	NCMG1	DCT-24	1
Zapotitlán de Vadillo	*A. rhodacantha*	Negro Cimarrón	NCZ1-3	MPDM-276	3
Cabo Corrientes	*A. rhodacantha*	Pencudo	DVDP6, DVDP1	MPDM-238	2
Cabo Corrientes	*A. aff. rhodacantha*	Verde	DVDV10	MPDM-237	1
San Gabriel (Apango)	*A. angustifolia*	wild	AP1-9	DCT-21, DCT-22	9
Milpillas (entronque Huaxtla)	*A. angustifolia*	wild	HUAX1-3	LMCC-150	3
San Cristobal de la Barranca	*A. angustifolia*	wild	CRIS2_1-16	LMCC-151	16
Sayula	*A. angustifolia*	wild	SAY1-7	DCT-20	7

The fourth cluster is the largest, comprising a cohesive group with highly related samples, mostly coming from plants growing in the wild. This group comprises a more genetically diverse set of samples from different regions. It is split in two major subgroups: 4a and 4b with excess of homozygotes and the highest values of Pi (Table 2). Cluster 4a includes all the wild plants sampled in San Cristobal de la Barranca (CRIS and HUAX coded samples), a region in the Northeast of Jalisco, geographically close to Tequila, the locality of origin of the blue agave. This cluster of wild plants also includes samples CIAH1 and CIAH2, that come from two plants classified as *A. tequilana* var. “Chato”. Cluster 4b is the most complex of all clusters. It includes several different landraces as well as plants from the wild. The most divergent sample within 4b is TET5, classified as “Lineño” by producers in Tetiapán (Zapotitlán, Southern Jalisco). Two subgroups in addition to TET5 can be found within 4b. The first includes all samples from different landraces in the North Coast, all collected in Cabo Corrientes: “Cenizo” (CJJ3, CJJ1), “Amarillo” (PDSA), and “Chico Aguiar” (samples coded as DVD, CHJJ, CJJ). The other subgroup includes all samples of plants from Southern Jalisco: samples collected in the wild from Apango (coded as AP) and Sayula (coded as SAY), several plants classified as “Cimarron” collected in Zapotitlán de Vadillo (PELA1, CLA1, CHLA1, TET2), Tolimán (TOL4), Tetiapán (TET2); “Negro Cimarrón” from Zapotitlán de Vadillo (NCZ) and Tolimán (NCMG1), “Cimarrón Verde” from Tolimán (CVMG2), “Azul Telcruz” from Zapotitlán de Vadillo (ATLA9) and Tolimán (SOMG1), “Lineño” from Tetiapán (TET6), “Ixtero verde” from Zapotitlán de Vadillo (IVLA3); as well as *A. tequilana* var. “Mano Larga” (C1ML1) and *A. tequilana* var. “Azul” (CIAT1-2).

The arrangement of our samples in the genetic clusters described above was also congruent with the results of the analyses of the genetic structure with ADMIXTURE. For these analyses, we selected *K* values from two to eight and ran ten repetitions with a random seed each. We observed that *K* = 5 had the lowest cross-validation error rate from the analysis. Then, we looked for the most representative repetitions with CLUMPAK. We selected the fifth repetition from *K* = 2 (7/10), the first repetition from *K* = 4 (8/10) and the eighth repetition from *K* = 5 (10/10) (see Figure 3). When *K* = 2, we observed that Clusters 1–4 also conform to different genetic entities, with the largest Cluster 4 as well as Cluster 3b, which include wild samples, having the highest diversity. Cluster 1, 2 and 3a show the most cohesiveness and less degrees of diversity; “Lineño” and “Ixtero Amarillo” landraces are mainly formed by one population, while “Amarillo” landraces form the counterpart. The supergroup shows a mixed ancestry. *K* = 4 was useful to separate the “Ixtero Amarillo”, “Amarillo” and “Lineño” landraces into three different ancestry groups. The fourth group was mainly present in the mixed supergroup and in the sister group of “Ixtero Amarillo”. When including a fifth ancestral group (*K* = 5), we noted *A. tequilana* and *A. rhodacantha* landraces such as “Chato”, “Azul” and “Chico Aguiar” are fully represented.

### 2.3. Morphological Differentiation

We gathered morphological data for 20 genotypes in our dataset (i.e., 23% of the total number of samples), focusing on several variables that have been useful to study traditional *Agave* landraces [24]. In total, we obtained measurements of 17 traits, which include Munsell leaf colors that were converted to xyY coordinates. After landrace identification by farmstead owners, all the 20 individuals were measured, most of them from Cabo Corrientes (North Coast) and four individuals from Zapotitlán de Vadillo (Southern Jalisco).

The measurements were used to build a heatmap (Figure 4a), which shows all retained morphological variables (only maximum leaf width was removed) in columns and samples in rows, ordered by its Euclidean distance and the UPGMA hierarchical clustering method. Although, some patters within variables related to different landraces could be seen, individuals included in each of them present morphological variation. As an example, the highest values of the three-color coordinates as well as the simple variables related to the terminal thorn are shown by the two individuals of “Ixtero Amarillo”, but the variables “terminal thorn length/terminal thorn base width” as well as “teeth length” present a variation among them. For this last variable, “Amarillo” individuals also show high variability.

Using our morphological measurements, we built a UPGMA dendrogram based on Euclidean distances. This dendrogram showed groups that are highly congruent with landrace determination as well as geography (Figure 4a). Three major groups with high bootstrap support were distinguished according to morphology. The first one and clearly separated from the pool of other individuals is the two “Ixtero Amarillo” plants from Zapotitlán de Vadillo. The second group is large and includes several landraces from Cabo Corrientes (“Amarillo”, “Verde”, and “Pencudo”), as well as two individuals measured in Zapotitlán de Vadillo identified as “Azul Telcruz” and “Ixtero Verde”, respectively. The last group is also large and includes plants from Cabo Corrientes identified as “Chico Aguiar”, “Pencudo”, and “Cenizo”.

The dendrogram built with morphological data was compared both visually and numerically (cophenetic and Baker correlation matrix indexes) with a genetic tree built with a reduced SNPs dataset that only includes the samples for which we had morphological measurements. This last dendrogram of genetic distances shows highly cohesive and supported grouping patterns that follow landrace classification. However, there is no full congruence amongst morphological and genetic dendrograms, as the congruence is more evident only in some groups (Figure 4b). Genetic and morphological distance matrices showed a Mantel correlation of 0.54 (*p*-value = 0.001) and the derived dendrograms a cophenetic and Baker matrix correlation of 0.54 and 0.7, respectively (with near 0 values meaning that the two trees are not statistically similar), showing high equivalence among them.

## 3. Discussion

As it happened with the history of tequila production in the region of west-central Mexico, the specific mezcaleras visited for this study began to be managed at least during the second half of the 18th century (Joya Hildegardo, *pers. com*). Even though producers identify several landraces, only few are commonly known and less ambiguously identified by most of them in each region, which might indicate the generational time elapsed since the start of management and the strength of their genetic identity. That is the case of landraces “Amarillo”, “Verde”, “Cenizo” and “Chico Aguiar” in the North Coast, as well as “Ixtero Amarillo”, “Ixtero Verde” and “Lineño” in the South. In contrast, other plants grown in these agroforest systems are classified using terms such as "Cimarrón”, “Sierreño” or “Barranqueño”. These terms refer to plants that farmers recently found in the wild and identified as good candidates to bring to their cultivars. This is the mechanism by which landraces probably start and then producers keep reproducing them asexually for generations. For example, the terms “Negro Cimarrón” or “Cimarrón Verde” indicate plants that have recently been brought from the wild or are under incipient management. Plants under these different stages of management suggested by their ethnobotanical denomination might show different levels of genetic diversity and differentiation [31]. Our results show more differentiation at the genomic level for the oldest and more extensively used landraces, and they also show the mixed ancestry for the most recent ones. In addition, we found that there is a morphological cohesiveness amongst the widely recognized landraces in comparison with the “Cimarrón”, “Sierreño” or “Barranqueño” plants, which might also be related to a particularly genetic blueprint. These last plants might show a relationship with individuals in the same geographic region and a mixed genetic component with wild plants.

### 3.1. Genetic Differentiation within the A. angustifolia Complex in Jalisco

This is the first study to report the genome wide characterization of traditional spirit-producing landraces included in the *A. angustifolia* complex in Jalisco, Mexico, together with morphological data for several samples, and the integration of these data with ethnobotanical information related to the management and domestication processes for spirit producing landraces suggested by their common names. After deep ethnobotanical explorations, Colunga-García Marín and Zizumbo-Villareal showed that southern Jalisco might be the nucleus of the greatest number and diversity of landraces used for the production of *Agave* spirits, particularly within the *A. angustifolia* complex [8]. However, *A. angustifolia* is used in other areas of the state to produce distillates, such as in the North Coast, where we are reporting new molecular and morphological data for local varieties.

The *A. angustifolia* complex comprises four species and five varieties, including three species used for mezcal production [25]. In this study we included, in addition to *A. angustifolia* landraces, two other species growing in Jalisco: *A. tequilana* and *A. rhodacantha*, although due to taxonomic difficulties in distinguishing the latter, we cautiously consider some of our specimens as *A. aff. rhodacantha*. Our approach, using GBS sequencing and SNPs calling, gave us access to a significantly larger number of loci to analyze (19,983 unlinked loci) in comparison with previous studies using traditional markers such as AFLPs or microsatellites [25,32]. Our results confirm the close relationship of *A tequilana*, *A. angustifolia* and *A. rhodacantha*, but do not show a clear differentiation of these three species in separate supported clusters. In contrast, our samples of *A. tequilana* and *A. rhodacantha* or *A. aff. rhodacantha* are mixed and intercalated amongst a diverse arrangement of *A. angustifolia* wild and landraces samples, suggesting the extensive gene flow and poor genetic limits existing amongst entities within the complex. However, a more conclusive assessment of the identity of these taxonomic units and landraces requires the inclusion of more specimens from different localities in the study, as well as morphological data for both vegetative as well as reproductive structures.

### 3.2. A. tequilana Genetic Background and Domestication

Previous information on the genetics within the *A. angustifolia* complex coming from AFLP studies showed the genetic differentiation but very close relationship of *A. angustifolia* and *A. tequilana*, even suggesting that *A. tequilana* is a type of *A. angustifolia* [33]. Additionally, using AFLPs, studies of cultivars in Oaxaca, Mexico, Rivera-Lugo and collaborators showed *A. tequilana* as a cohesive group closely related to *A. angustifolia* var. “Espadin” [25]. These results are congruent with the taxonomical and ethnobotanical literature, which suggest the origin of the cultivar from wild populations of *A. angustifolia* growing on the semi-arid slopes in Cocula and Tecolotlán in Central Jalisco [2,8].

Given its importance in tequila production, *A. tequilana* and the suggested landraces have been studied in more depth than *A. angustifolia*. The study of the genetics of *A. tequilana* with traditional genome-wide genetic techniques such as AFLPs and RAPDs were inconclusive, showing either low or significant diversity depending on the technique used [17,32]. These molecular markers have a high mutation rate and can change from one generation to the next [33], or even in between bulbils produced asexually from the same mother plant [34]. By using microsatellites, Trejo et al. [26] studied several cultivars and populations in the *A. angustifolia* complex in Jalisco, finding that *A. tequilana* var “Azul” samples were more closely related to the *A. angustifolia* populations from southern Jalisco. In addition, they found that the *A. tequilana* var “Sigüin” and *A. tequilana* var “Chato” as well as the landrace “Ixtero Amarillo” are closely related.

For our study, we included in our analyses two accessions corresponding to the *A. tequilana* var. “Chato” (CIAH-1, CIAH-2), and two accessions of *A. tequilana* var. “Azul” (CIAT-1, CIAT-2, TOL13), as well as one accession of *A. tequilana* var. “Mano Larga” (C1ML-1). Interestingly, and similarly to Trejo et al. [26], our results suggest an independent domestication of these three varieties from different wild *A. angustifolia* backgrounds. We also found more genetic variation in these *A. tequilana* landraces than previously thought, which deserves more in-depth studies. Our results indicate that the *A. tequilana* var. “Chato” in the cluster 4a (Figure 3) emerged from populations of *A. angustifolia* in San Cristobal de la Barranca and Milpillas (CRIS and HUAX coded samples), a region located to the north of Guadalajara city, geographically close to Tequila, the locality of origin of the blue agave. On the other hand, our samples of *A. tequilana* var. “Azul” and *A. tequilana* var. “Mano Larga” both group in a totally different cluster, 4b, which is the largest in our study and includes several landraces and wild plants collected in the south of Jalisco, suggesting that these two other *A. tequilana* landraces might have their origins in that region.

### 3.3. Traditional Landraces and Their Morphological Identity

Agave spirits are produced all over Mexico, applying both industrialized or semi-industrialized as well as traditional and artisanal methods using different species of *Agave*. In most localities, particularly where spirits are distilled traditionally, producers still recur to wild plants and follow locally developed non-industrialized methods. In other regions where the production of spirits has deeper heritage and history, families of farm holders and producers inherit a tradition of cultivating and propagating plants that were initially brought from the wild generations ago. It has been suggested that landraces such as “Ixtero Verde”, “Ixtero Amarillo”, “Cenizo” or “Lineño” have been cultivated in the region from at least 100–150 years [24].

Some inherent biological characteristics of *Agave* spirit producing crops provide particularities to their domestication processes and establishment of landraces. The long life- cycles of *Agave* plants (around 6 to 10 years or more), and the fact that reproductive structures that allow sexual crossing are cut before maturity during their cultivation, make breeding practices practically inexistent. Plants recognized as landraces might come from one or a few genetic lines of plants found originally in the wild, and then propagated over and over asexually through clonal rhizome plantlets or bulbils. Thus, we expect the genetic variation within landraces to be significantly reduced. We could also expect that morphological characteristics are highly consistent; however, the recognition and naming of landraces by producers involves a social process of acceptance of the group of morphological traits, particularly if it occurs amongst different farm holders over larger geographic areas. For example, although most landraces are cultivated by single producers, others such as “Lineño” and “Ixtero Amarillo” are more widely recognized and distributed, suggesting a longer history of propagation and cultivation [27].

In the case of the *A. angustifolia* complex, the morphological characteristics of several landraces have been studied extensively in Jalisco by Vargas-Ponce and collaborators. These authors report the presence of at least 24 traditional landraces cultivated in the state, most of which are grown in Southern Jalisco. In contrast, Central Jalisco is currently dominated by monovarietal plantations of *A. tequilana* var “Azul” for tequila production, and traditional landraces have become very scarce [4,8]. Our study extends to the analyses of landraces in another region in Jalisco, the North Coast, where the spirit produced is often named “Raicilla”. To our knowledge, this is the first time that these landraces are studied. For our morphological analyses, we used the same variables as Vargas-Ponce et al. [24]. With this set of morphological variables, Vargas-Ponce et al. [24] found that the landraces in southern Jalisco are already morphologically differentiated. Barrientos-Rivera et al. [35] study *A. angustifolia* landraces in Guerrero, Mexico using a similar set of variables, and they were also able to separate the individuals belonging to populations of the “Sacatoro” landrace from the population of the “Espadín” landrace included in their study. With this set of morphological variables, we were also able to confirm that the studied landraces already show morphological differentiation and cohesiveness at this organismic level, showing the adequacy and sufficiency of the morphological variables selected to study the differentiation of landraces in the *A. angustifolia* complex in Jalisco, as well as in other regions of Mexico.

Although with a limited sampling, our analyses of morphological data clearly differentiated landraces that form strongly supported clusters: “Cenizo”, and “Chico Aguiar” from the North Coast and “Ixtero Amarillo” from the South (Figure 4b). The plants classified as “Amarillo” landraces show the highest diversity, and they form a large cluster such as “Pencudo” from the North Coast, but also “Azul Telcruz” and “Ixtero Verde” from Southern Jalisco. Unfortunately, for these last landraces we have very limited sampling, and the inclusion of more plants coming from different farms and regions is necessary to better assess both their morphological and genetic establishment.

### 3.4. Genetic Differentiation of Landraces

Our sampling design allowed for the collection and comparison of wild plants and plants growing under different levels of domestication and management (tolerated, encouraged and cultivated) at the genomic level. We generated a set of approximately 20K high-quality SNPs by RAD sequencing. The restriction site associated (RAD) DNA sequencing is one of the most common methods to genotype by sequence representative loci from all the genomes in plant populations [36]. With our dataset, we were able to characterize the genetic differentiation and variation in samples of plants within the *A. angustifolia* complex assigned to several autochthonous landraces under different levels of acceptance and width of use in Jalisco. We also included a significant number of wild plants from the same geographical region, which allowed us to have a better picture of the domestication process in the wild genetic context. With the genomic information obtained in this study, we were able to fine discriminate among producer’s varieties, showing that several landraces have a genomic basis. The use, preservation and promotion of these traditional *Agave* landraces has been proposed as an alternative to reduce the impact of the extensive monovarietal cultivation of *A. tequilana* var. “Azul” [24].

Archaeological evidence shows that *Agave* species and landraces have been used in the Jalisco and surrounding regions for at least 2500 years, and that they have been very important in both daily and ceremonial life [15,37]. For example, the word “Ixtero” gives the name to one of the most representative and ancient landraces in southern Jalisco: “Ixtero Amarillo” [37]. The word “Ixtero” is derived from the Náhuatl language (Mexican original native prehispanic language) “ichtli” or “ixtl”, name of the fiber produced with *Agave* spp., and refers to the first use of this specific landrace [37]. According to Colunga-GarcíaMarín and Zizumbo-Villarreal [8], with the introduction of the distillation process, the agaves, first selected for food or their fibers, were then submitted to a second selection pressure to provide the best germplasm for the production of distilled spirits [8]. In this context, “Ixtero Amarillo” has been cultivated mostly by shoots for probably more than 200 generations of agave plants (considering 12 years to mature on average, Miguel Partida, *pers. com*).

Based on genomic data, our results show a clear differentiation in the landraces that are best recognized by farmers: two in Southern Jalisco (“Ixtero Amarillo” and “Lineño”) and one in the North Coast (“Amarillo”), that conform separate supported clusters (Cluster 1, 2 and 3), even when the plants were sampled from different parcels or farms. These groups, particularly “Amarillo”, show small Ho values (Table 2), indicating the low genetic diversity expected particularly for a group that has been reproduced vegetatively for generations. However, the “Ixtero Amarillo” and “Lineño” groups (Clusters 2 and 3) show higher genetic diversity, and in our dendrogram, they are genetically close to plants recognized under landrace names such as “Barranqueño” or “Cimarrón”. These terms are relatively generic and used by farmers to refer to plants recently brought from the wild. Thus, both the ethnobotanical information contained on the designation of landraces and the genetics are congruent with a still incipient process of domestication and homogenization of these landraces. Unfortunately, we only have morphological information for two “Ixtero Amarillo” plants and none for the “Lineño” to allow for the confirmation of this intermediate level of domestication at the morphological level.

The “Amarillo” landrace, the most clearly differentiated landrace in our study both genetically and morphologically, sustains 95% of the Raicilla spirit production on the North Coast (Pedro Jiménez, *pers. com*). In contrast with the scenario found in Southern Jalisco, there are no historical or anthropological studies on the management of *Agave* in the Cabo Corrientes municipality. However, producers share similar traditions with the ones in Southern Jalisco, such as the Filipino distillation method, which incorporates stills made with hollow trunks. In both regions, producers refer to their spirit factories as “taverns” and the fermented beverage that they distillate as “tuba”. Thus, the clear genomic differentiation of “Amarillo” and common cultural traits related to the elaboration of spirits between the North Coast and Southern Jalisco suggest a similar human selection process in both regions and are congruent with the hypothesis of Bruman [6,7], suggesting a quick spread of Filipino distillation techniques from Southern Jalisco to the North.

The “Lineño” landrace is better represented in Tolimán, the neighboring municipality of Zapotitlán de Vadillo, but it is also frequent in other municipalities not included in this study (e.g., Tonaya, [8]). This landrace is recognized as one of the highest yielders, with a shorter production cycle and great capacity to produce shoots. It is, together with “Ixtero Amarillo”, the most mentioned and recognized by all interviewed farmers in both municipalities, and it is currently the most common in Tolimán, where it is grown in monocultures with a high density of plants, sometimes together with *A. tequilana*, and also reproduced mostly by bulbs.

Finally, samples that came from landraces used with less frequency by producers are all grouped within the large Cluster 4 (Figure 3, left), which are closely related to most of our wild samples. This lack of differentiation between cultivated and wild populations was already noticed in the sampling of *A. angustifolia* var. *pacifica* in Sonora, the north of Mexico [31]. This is congruent with affirmations from the oldest farmers interviewed, who narrate that they used to have an active and constant practice of germplasm selection from the surrounding hills. However, this is not the case for recent generations of farmers, who seek more established landraces.

Selection criteria for spirit production vary between producers. Some valuable traits are linked to the production of larger, heavier heads, high sugar content, resistance to pests, diseases and foragers, reproductive precociousness, flavor, or meeting commercial criteria [24]. The generation of SNPs sites allowed us to evaluate the genetic variability and differentiation in these landraces in comparison with wild plants globally along the genome and not only in a few loci. The future availability of high-quality reference genomes for major landraces in the *A. angustifolia* complex, together with a more extensive study of landraces with an increased sampling, will allow the detection of loci of interest related to the desired valuable traits, as well as a correlation to regional abiotic conditions.

## 4. Conclusions

In this study, we report for the first time a genome-wide characterization of Jalisco spirit producing *A. angustifolia* autochthonous landraces using a GBS approach. This geographic area is highly relevant because it holds the greatest diversity of landraces in the *A. angustifolia* complex; it is also the putative center of origin of agave distillation in general, and the domestication of the blue agave tequila plant in particular. Our approach, using GBS sequencing and SNPs calling, gave us access to a significantly larger number of loci to analyze in comparison with previous studies using traditional markers such as AFLPs or microsatellites. Our data show that the more extensively used and widely recognized traditional landraces such as “Ixtero”, “Lineño” or “Amarillo” have an important degree of differentiation, both at the genetic and morphological level. However, the differentiation is not absolute and there is a very close relationship with wild genotypes. This might be the result of the extremely recent management and domestication of landraces, together with the important use of clonal propagation. We document the genetic basis of the domestication process in the area, where farm holders recur to wild plants to select and propagate new local varieties that reduce their genetic diversity after several generations of clonal propagation. This method promotes the acquisition of locally adapted germplasm and the diversification of available landraces.

The information we provide enables a better characterization of traditional landraces in the state of Jalisco, Mexico. We hope that it helps in technologizing their use and management and provides tools to foment and protect traditional landraces and farming systems to preserve the agrobiodiversity they hold.

## 5. Materials and Methods

### 5.1. Sample Collection

In total, we collected 87 samples in the state of Jalisco, Mexico during 2020 and 2021 (Figure 1). We visited twelve mezcaleras in three municipalities: Cabo Corrientes (four localities) on the North Coast, and Zapotitlán de Vadillo (two localities) and Tolimán (one locality) in Southern Jalisco. In addition, we collected several wild populations around both North and South areas of study, as well as central Jalisco near Tequila, a locality of putative origin of the *A. tequilana* var. “Azul”. When sampling cultivated plants, producers were interviewed to document the landraces they use and how they recognize them. With their help, we selected several mature plants per landrace and took morphometric measurements. Our pool of samples includes 14 landraces from both *A. angustifolia* and *A. rhodacantha* (Table 1, Figure 2). Additionally, we included five *A. tequilana* samples of plants growing on the CINVESTAV Irapuato botanical garden whose origin are private blue agave destined plantations in Tequila, Jalisco (Simpson J. *pers. com*).

In total, twelve producers were visited and interviewed in a semi-structured way to document the landraces they use and how they recognize them. With their help, 10 mature plants per accession; that is, near to the reproductive state; were selected. Though most of the plants are vegetatively propagated, producers usually obtain them from different mother plants on the same farm. Efforts were made to have each variety represented by more than one accession in order to avoid clones, except in the cases where varieties are maintained by a single producer. To assess the wild genetic background of the landraces studied, we sampled *A. angustifolia* plants growing in the wild. In this case, plants sampled were located far enough from farmsteads, far from residential or agricultural properties, and evidently growing under isolation and not under cultivation. When wild plants were isolated, we collected only tissue samples for DNA extraction. When wild plants were found in colonies or populations, ten individuals separated by more than 10m were sampled and an herbarium voucher specimen was prepared (Table 1).

### 5.2. Morphological Data and Analyses

We gathered morphological information for 20 samples within our genetic dataset (i.e., 23%). The morphological variables measured have previously shown to be useful in the study and characterization of traditional agave landraces [24]: plant length (cm), leaf length (cm), maximum leaf width (cm), leaf width at the middle of the leaf (cm), terminal thorn length (cm), terminal thorn base width (cm), number of lateral teeth, distance between teeth (cm) and teeth length (cm). Munsell leaf colors were also recorded and converted to xyY coordinates. In addition, the other 5 variables were calculated as combinations of the previous ones: distance between teeth/leaf length, number of lateral teeth/leaf length, terminal thorn length/terminal thorn base width, leaf length/terminal thorn length and leaf length/leaf width at middle. Variables highly correlated (>0.9 Pearson coefficient) were removed. All the measurements were taken from plants growing in mezcalera agroforest producing systems from Cabo Corrientes and Zapotitlán de Vadillo, thanks to the owner’s collaboration. To explore the congruence amongst morphological features and landrace information provided by producers, a normalized heatmap in the range (0–1) was constructed in R with BBmisc [38], palette [39] and stats (R core) packages, and the resulting UPGMA Euclidean distance dendrogram was compared both visually and numerically (cophenetic and Baker correlation matrix indices) with a genetic tree built just for the plants, to which we had morphological data using R package dendextend [40]. The genetic tree was also built with UPGMA clustering using the Euclidean distances from the first three principal components of the PCA obtained with the unlinked SNP. To evaluate support, both morphological and genetic trees were bootstrapped 100 times using the R package ape [41]. A mantel test was performed among the morphological and genetic pairwise distance matrices (vegan R package, [42]). All calculations were performed in R using the following packages: vcfR [43], VariantAnnotation [44], BBmisc [38], adegenet [45], ape [41], paletteer [39], dendextend [40] and vegan [42].

### 5.3. DNA Sequencing and Bioinformatic Analyses

Tissue samples of plants collected in the field were either immediately frozen on liquid N or maintained at 4 °C for 24 to 36 h and then transferred to a −70 °C storage. Approximately 300 mg of each sample was grounded using liquid nitrogen and high-molecular-weight DNA was extracted (DNeasy Plant 96 QUIAGEN^®^ Kit). Genomic DNA extracted was checked for degradation using a 1% agarose gel electrophoresis, and the quality was determined using NanoDrop 8000 (Thermo Scientific (Waltham, MA, USA)). Samples were standardized and library preparation was performed at the Genomics Services Laboratory in Langebio, Mexico. Samples were double digested using the restriction enzymes BglII y DdeI and sequenced using the NovaSeq Illumina platform for 5 million single end 100bp reads per sample (1 × 100).

Per-base and per-read quality score statistics were calculated for each fastq file through FastQC v0.11.1 [46] and MultiQC v1.0 [47]. After demultiplexing, mean quality scores for all libraries were Q ≥ 50 and no adapters were present. The unpublished *Agave tequilana* reference genome was provided by LANGEBIO CINVESTAV Irapuato (Irapuato, Mexico) (*in prep*) and was used as a reference to align sequenced reads using the default algorithm of the Burrows–Wheeler Aligner v0.7.15 (bwa mem [48]), and the BAM file was sorted and indexed using SAMTOOLS v1.9 [49]. These BAM files were used for the SNPs calling process and assessed for sequencing depth using SAMTOOLS (samtools depth). Loci were identified using the *gstacks* pipeline with default parameters in STACKS v2.3e [50].

### 5.4. Population Genetics and Statistical Analyses

Population genetics statistics were computed using the populations program in STACKS v2.3e [50] with the following arguments: 80 percent of individuals within and across populations were required to process a locus, a minimum number of 2 populations a locus must be present to process a locus, a minimum minor allele frequency of 0.1 and a minimum allele count of 2 was required to process a nucleotide site at a locus (-p 2 -r 0.8 -R 0.8--min-maf 0.1--min-mac 2 -H). Finally, we removed all SNPs that did not pass these filters. Population genetic parameter estimations (*H_e_*, *F_ST_* and ***π***) were calculated using 1000 bootstrap repetitions, *p*-value *F_ST_
*correction and kernel smoothing (--fst_correction ‘*p*-value’--bootstrap 1000--fstats--k).

To select representative unlinked SNPs, the original dataset was pruned by linkage disequilibrium (LD) using PLINK v1.9 [51] with standard parameters (--indep-pairwise 50 10 0.1). PLINK was also used with this dataset to calculate Nei’s genetic distances (--ibs), to perform identity-by-descent analysis (--genome) and to calculate the principal component analysis and genetic distances. We selected the eigenvectors from the first three components to build a hierarchical clustering (hclust (); method = “average”) within R. Finally, we carried out ADMIXTURE analyses using K values ranging from 2 to 8, ten random seeds and 200 bootstraps per K value (ADMIXTURE v1.3.0 [52]) and selected the best K values, as well as identify the most representative repetitions with the lowest error rates using the CLUMPAK online pipeline (http://clumpak.tau.ac.il/; accessed on 14 May 2022 [53]). Genetic diversity indexes for the different groups were calculated as follow for two partitions of the data set; individuals of the groups 1, 2, 3, 4 and groups 1, 2, 3a, 3b, 4a, 4b obtained after analyzing the PCA based dendrogram. Expected heterocigosity (*He*), observed Heterocigosity (*Ho*) and inbreeding coeffient (*Fis*) were calculated with the R packages hierfstat [54] and adegenet [45,55]. Nei Pi nucleotide diversity was calculated with the R package PopGenome [56].

## Figures and Tables

**Figure 1 plants-11-02274-f001:**
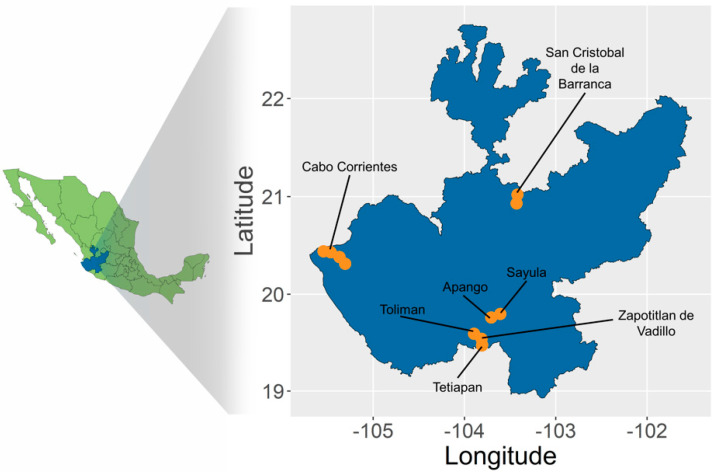
Area of study (Jalisco, Mexico) and localities where samples were collected.

**Figure 2 plants-11-02274-f002:**
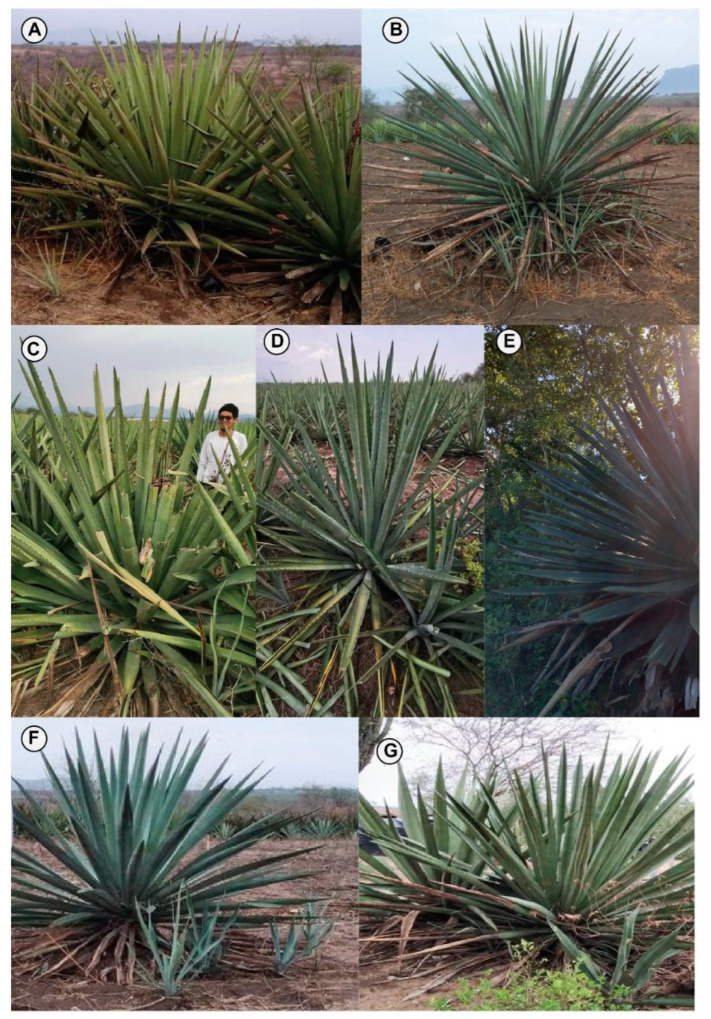
*A. angustifolia* landraces of Jalisco. (**A**) Ixtero Amarillo, (**B**) Ixtero Verde, (**C**) Lineño, (**D**) Barranqueño, (**E**) Cenizo, (**F**) Azul Telcruz, (**G**) Cimarrón.

**Figure 3 plants-11-02274-f003:**
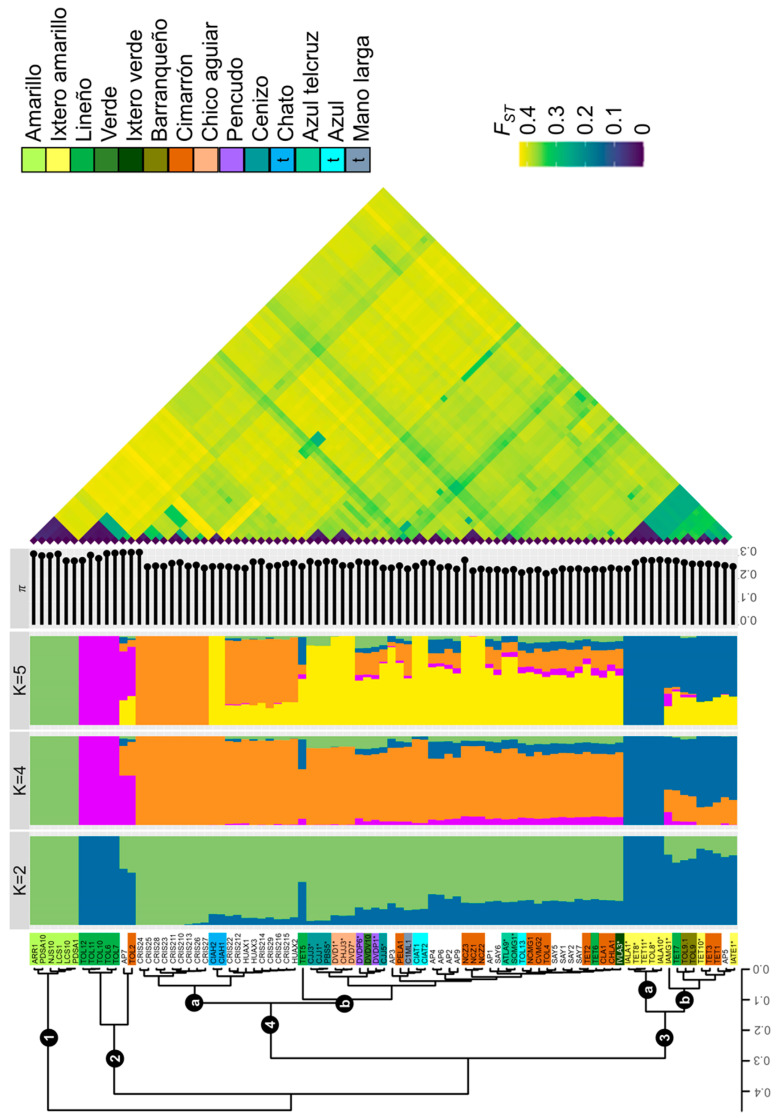
UPGMA, ADMIXTURE, ***π***, and *Fst* analyses results. Landraces are color coded. t = *A. tequilana*, * = *A. rhodacantha* or *A. aff. rhodacantha*.

**Figure 4 plants-11-02274-f004:**
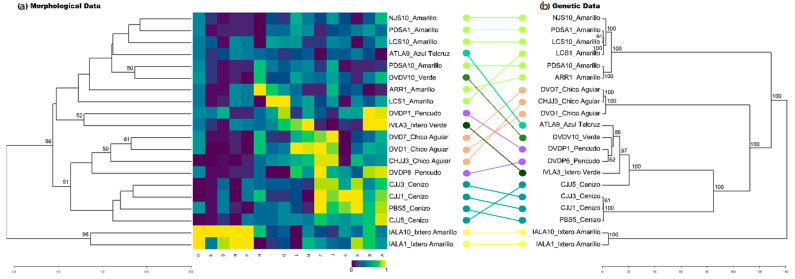
Dendrograms obtained with the morphological (**a**) and genetic (**b**) 23% subset data. (**a**) Morphological dendrogram and heatmap with normalized values to a range 0–1. Variables are: (A) plant length, (B) leaf length, (C) leaf width at middle, (D) terminal thorn length, (E) terminal thorn base width, (F) number of lateral teeth, (G) distance between teeth, (H) teeth length, (I) Distance between teeth/leaf length, (J) number of lateral teeth/leaf length, (K) terminal thorn length/terminal thorn base width, (L) leaf length/terminal thorn length, (M) leaf length/leaf width at middle and Munsell coordinates (N) xyY.x, (O) xyY.y and (P) xyY.Y.

**Table 2 plants-11-02274-t002:** Genetic diversity indexes for the Clusters obtained with the PCA based dendrogram. Expected heterocigosity (He), observed Heterocigosity (Ho) and, inbreeding coefficient (Fis) and nucleotide diversity (pi).

Cluster	n	Ho	He	Fis	pi
1	6	0.2884	0.1546	−0.8011	102.7727
2	7	0.3017	0.2155	−0.2893	125.0220
3	14	0.3144	0.2667	−0.1198	151.5476
4	60	0.2850	0.3337	0.1543	182.6654
3a	5	0.3204	0.1708	−0.8420	113.8000
3b	9	0.3109	0.2946	−0.0520	165.0458
4a	20	0.2711	0.3157	0.1326	174.0885
4b	40	0.2925	0.3300	0.1222	180.4139

## Data Availability

The manuscript describing the *A. tequilana* draft genome is still in preparation. However, access and permission to access the data before publication can be arranged by directly contacting Dr. Alfredo Herrera-Estrella (alfredo.herrera@cinvestav.mx) or Dr. Selene Fernández-Valverde (selene.valverde@cinvestav.mx).

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
