# Peer review of "Genomic and Morphological Differentiation of Spirit Producing Agave angustifolia Traditional Landraces Cultivated in Jalisco, Mexico"

_plants, 2022, doi:10.3390/plants11172274_

Round 1

Reviewer 1 Report

General comments

The article is quite interesting. However, in my opinion the introduction is extremely long, it should be shortened. The final conclusions should be a little clearer

Specific comments:

Line 60: DOT. It should be explained what it means the first time it appears

Line 107: The purpose of the study should be clearly explained.

Lines 121-123: This is a conclusion, I shouldn't be here

Lines 127 – 141: This is material and methods not results.

Author Response

Dear Reviewer 1, 

We are very thankful for your comments and suggestions, since they contributed to significantly improve our manuscript. Below we explain the corrections we made following your advice:

*the introduction is extremely long, it should be shortened.*

We performed the following changes to reduce the length of the Introduction:

  • Paragraph 2 and paragraph 3 in original manuscript were merged and shortened in a single paragraph in the new version of the manuscript.
  • Paragraph 4 and paragraph 5 in original manuscript were also merged and shortened in a single paragraph
  • We cut unnecessary information along the Introduction

With the changes made, we reduced the original length of the Introduction (1100 words) to 868 words. We hope this shortened version fulfills an acceptable length. However, we are willing to reduce more if Reviewer considers it is needed.  

For the specific comments:

*Line 60: DOT. It should be explained what it means the first time it appears*

This has been corrected in the new version of the manuscript

*Line 107: The purpose of the study should be clearly explained*

We modified the writing to state the purposes of the study more clearly and at the beginning of the paragraph rather than at the end

*Lines 121-123: This is a conclusion, it shouldn't be here*

We removed these lines from the Introduction and relocated them in the new Conclusions section.

*Lines 127 – 141: This is material and methods not results.*

The paragraph was moved to the Materials and Methods section

*The final conclusions should be a little clearer*

In the new version of the manuscript, we included a Conclusions section, that we hope makes conclusions clear enough. We are happy to modify or improve the writing if considered.

Reviewer 2 Report

The manuscript is well-written, I recommend to accept it.

Only one little comment:

In the title (line 4) - maybe the full name of the species...??

Author Response

Dear Reviewer 2, 

We are very thankful for your comments and suggestions, since they contributed to significantly improve our manuscript. Below we explain the corrections we made following your advice:

In the title (line 4) - maybe the full name of the species...??

In the new manuscript, the title was modified to write the whole word: Agave, instead of A.